# Mycorrhizas drive the evolution of plant adaptation to drought

Marco Cosme [1 ✉]

Plant adaptation to drought facilitates major ecological transitions, and will likely play a vital role under looming climate change. Mycorrhizas, i.e. strategic associations between plant roots and soil-borne symbiotic fungi, can exert strong influence on the tolerance to drought of extant plants. Here, I show how mycorrhizal strategy and drought adaptation have been shaping one another throughout the course of plant evolution. To characterize the evolutions of both plant characters, I applied a phylogenetic comparative method using data of 1,638 extant species globally distributed. The detected correlated evolution unveiled gains and losses of drought tolerance occurring at faster rates in lineages with ecto- or ericoid mycorrhizas, which were on average about 15 and 300 times faster than in lineages with the arbuscular mycorrhizal and naked root (non-mycorrhizal alone or with facultatively arbuscular mycorrhizal) strategy, respectively. My study suggests that mycorrhizas can play a key facilitator role in the evolutionary processes of plant adaptation to critical changes in water availability across global climates.

[1] Mycology, Earth and Life Institute, Université Catholique de Louvain, Croix du sud 2, 1348 Louvain-la-Neuve, Belgium. ✉email: marcosmeweb@gmail.com

The adaptation of plants to contrasting gradients of water availability has facilitated major ecological transitions on Earth, such as the migration from exclusively aquatic environments to terrestrial habitats[1,2], and the subsequent expansions across nearly all land surfaces[3], of which about half are currently susceptible to droughts[4]. These worldwide evolutionary adaptations in plants have coexisted for millions of years with persistent mutualistic strategies between plant roots and mycorrhizal fungi[5,6], that exert a strong influence on drought tolerance and survival of extant plants[7,8]. However, how drought adaptation and mycorrhizal strategy influence one another throughout the course of plant evolution is unknown.

Droughts occur in nearly all climatic zones, from high to low rainfall areas, and result primarily from a reduction in precipitation below the normal levels over extended periods of time, such as a season or a year[4]. Paleorecords suggest that droughts led to significant vegetation shifts in a distant past[9,10]. In recent years, droughts have been experienced with higher peaks and severity due to climate change and have already caused a few vegetation shifts in different climates and vegetation types[4,11]. Current projections indicate that future global warming will lead to the increased manifestation of droughts in regions where drought intensification does not yet occur[12–14]. This will likely alter the regimes of natural selection and contribute to the loss of global phylogenetic diversity[15].

Plants, as the main primary producers that fixate carbon and provide shelter and food for a myriad of organisms, are essential to most land ecosystems across climatic zones, and are on the frontline of perilous droughts. Plants can either survive or succumb to drought depending on whether they hold drought tolerance mechanisms[14,16,17]. Yet, our current understanding of how drought tolerance evolves in plants is extremely limited. Only one study has investigated this across vascular plants based on 178 extant species, of which more than two-thirds are domesticated[17]. Hence, data on a large diversity of species living in nature across the globe is critically needed to narrow down uncertainties when characterizing the evolution of plant adaptation to drought in terrestrial ecosystems.

Mycorrhizas are known to improve stress tolerance and survival of plants during drought[7,8,18–21]. This predominant positive perception would suggest that mycorrhizas might accelerate the evolutionary gains of drought tolerance in plants due to the transgenerational success of mycorrhizal plants that survive repeated droughts, compared with that of the less successful non-mycorrhizal counterparts. However, mycorrhizas can also have neutral or detrimental impacts on plants during drought[8]. In addition, there are different classes of mycorrhizas that can influence drought tolerance in plants, namely arbuscular mycorrhiza, ectomycorrhiza, and ericoid mycorrhiza[18–20]. Thus, plants with different mycorrhizal strategies might experience differently the evolutionary pressures caused by critical environmental changes in precipitation regimes, and this might also differ from that of plants that either do not form mycorrhizas or form it to a much lesser extent[5,22,23].

Here, I describe 325 My of the evolution of drought adaptation and mycorrhizal strategy in land plants, based on a global sample of 1638 extant species (angiosperms and gymnosperms). To this end, species' data and phylogenies were compiled from several large-scale databases[24–28]. The trait drought adaptation was simplified into a binary variable to streamline model parameterization, with species having either a "tolerant" or "sensitive" state, following a previous approach[17]. Likewise, the trait mycorrhizal strategy was simplified into a ternary variable using a previously described approach[29]. Central to this approach is the observation that plants have evolved functionally different mycorrhizal strategies to acquire soil resources: (i) by scavenging

plant-available nutrients primarily in symbiosis with arbuscular mycorrhizal (AM) fungi[29,30] (hereafter AM state); (ii) by mining organic-bound resources mainly in symbiosis with either ecto-mycorrhizal or ericoid mycorrhizal fungi[29,31,32] (hereafter combined as EEM state); (iii) or by taking up resources mostly via the absorptive surface of their own naked roots[5,22,23,29] (hereafter NR state). Overall, this study includes 628 genera, 151 families, and 50 orders, representing a large diversity of terrestrial plants currently living in nature across the globe. With these data, I ran a series of recently developed hidden Markov models (HMMs) of varying evolutionary complexity, which arguably increase the accuracy along deep phylogenies[33], with the objective to test the hypotheses that: (1) the evolutions of drought adaptation and mycorrhizal strategy depend on each other and (2) mycorrhizas markedly influence the speed of evolution of drought adaptation in land plants. Drought adaptation is defined here as adaptative changes from drought sensitivity to drought tolerance as well as from drought tolerance to drought sensitivity. Hence, it reflects plant adaptation to drier habitats as much as plant adaptation to wetter habitats, in which drought sensitivity is an advantage for the plants. Finally, I analyzed the sensitivity of the main conclusions in relation to data and parameter estimate uncertainties.

My analysis indicates that the evolutions of mycorrhizal strategy and drought adaptation in land plants depend on each other. Plant lineages harboring consistent mycorrhizal symbioses associate with faster evolutionary shifts in drought adaptation. These results provide a quantitative demonstration that the suit of host-associated microbes can play a key role in the evolution of host adaption to critical environmental changes in water availability.

## Results

**The evolutions of mycorrhizal strategy and drought adaptation in plants depend on each other.** My global analysis on angiosperms and gymnosperms (Fig. 1) provides strong statistical support for a dependent evolution between drought adaptation and mycorrhizal strategy in land plants, as indicated by the goodness of fit among multiple HMMs with either an independent or dependent mode of evolution (Table 1)[33], supporting my first hypothesis. This signifies that, throughout the course of plant evolution, the rate of change in drought adaptation—i.e., evolutionary shifts between the drought-sensitive and -tolerant states—in a given lineage depends on the mycorrhizal strategy formed by that lineage—i.e., whether AM, EEM, or NR state—and the rate of shifts among mycorrhizal strategy depends on the lineage's adaptation to drought (Table 2). The robustness of this conclusion was confirmed against three main sources of data uncertainty, i.e. drought adaptation data, mycorrhizal strategy data, and phylogenetic tree backbone, which varied among six partially different dataset versions analyzed (Tables 1, 3). Furthermore, the multiple HMMs tested also differed from each other in terms of hidden rate categories (from one to three) and evolutionary model structure (from fully homogenous to fully heterogenous) (see Methods). The best-fitted HMMs revealed that this correlated evolution was consistently influenced by an unobserved (hidden) phylogenetic factor with two state levels (Table 1), which has either promoted or constrained the speed of evolution of the observed characters' drought adaptation and mycorrhizal strategy (Table 2). This means that, when inferring the transition rates among the observed states, the models took into consideration site-specific rate heterogeneities along the phylogeny, which may have resulted, for instance, from diversification events. For each of the six dataset versions analyzed, the model with two hidden rate categories was better fitted than the three-rate-category equivalent (Table 1). This indicates that a simple two-class

**a**

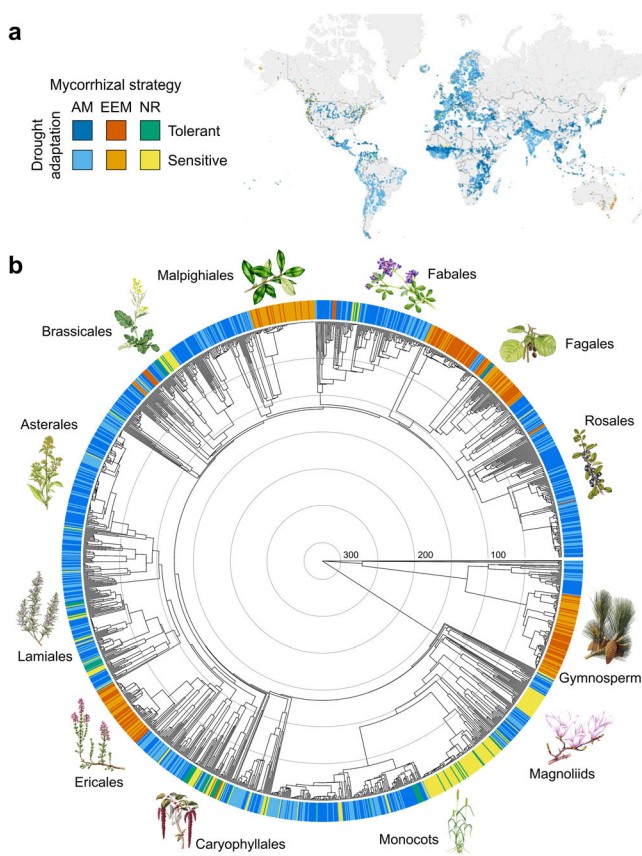

**b**

**Fig. 1 Global distribution and phylogeny of plants showing the mycorrhizal strategies and drought adaptations. a** Global geographical occurrences colored according to the mycorrhizal strategy and drought adaptation states of 65% of the plant species included in the phylogenetic tree shown in **b**. For individual geographical occurrences, see Supplementary Data 2 (found at https://doi.org/10.5061/dryad.3ffbg79nx). **b** Time-calibrated (mega-annum; Ma) phylogenetic tree of 1638 plant species (angiosperms and gymnosperms) with a colored band at the tips indicating the mycorrhizal strategy and drought adaptation state of each species as described in the legend shown in **a** and assigned according to dataset v1 (Methods). AM arbuscular mycorrhizal state, EEM combined ectomycorrhizal and ericoid mycorrhizal state, NR naked root state, i.e., non-mycorrhizal together with facultatively AM plants (dataset v1). The files of this and of the phylogenetic trees of the other dataset versions, with legible species labels, as well as the respective data frames with mycorrhizal strategy and drought adaptation states, are provided in Supplementary Data 1 (found at https://doi.org/10.5061/dryad.3ffbg79nx). The concentric circles inside the phylogenetic tree indicate 50 Ma. Botanical illustrations credits: Lizzie Harper/Science Photo Library (Brassicales, Ericales, Fabales, Magnoliids, Malpighiales, Monocots, and Rosales), Luis Montanya/Marta Montanya/Science Photo Library (Asterales, Fagales, and Lamiales), and Natural History Museum, London/Science Photo Library (Caryophyllales and Gymnosperms).

organization was more suitable to describe the detected hidden rate heterogeneities. Furthermore, this correlated evolution across the plant phylogeny was best characterized by a full or partial heterogeneous speed of evolution, which depended largely on the size of the dataset analyzed (Tables 1, 4). Finally, the calculated phylogenetic imbalance ratio of each dataset version was at least 20 times below the maximum recommended threshold[34] (Table 5), which ruled out issues of evolutionary sample size and phylogenetic imbalance[34–36], and provided support for the adequate detection of correlation evolution.

**Table 1 Model rankings from the maximum-likelihood analysis of the evolutionary relationship between drought adaptation and mycorrhizal strategy for each dataset version analyzed.**

| Evolution mode | Hidden rates | Model structure | k rates | Dataset v1 AICc | Dataset v1 AICcWt | Dataset v2 AICc | Dataset v2 AICcWt | Dataset v3 AICc | Dataset v3 AICcWt | Dataset v4 AICc | Dataset v4 AICcWt | Dataset v5 AICc | Dataset v5 AICcWt | Dataset v6 AICc | Dataset v6 AICcWt |
|---|---|---|---|---|---|---|---|---|---|---|---|---|---|---|---|
| Independent | 1 | ER | 2 | 2846.1 | <0.01 | 2846.1 | <0.01 | 2808.3 | <0.01 | 2808.3 | <0.01 | 1308.3 | <0.01 | 1308.2 | <0.01 |
| Independent | 1 | SYM | 4 | 2823.2 | <0.01 | 2823.2 | <0.01 | 2811.0 | <0.01 | 2785.4 | <0.01 | 1288.4 | <0.01 | 1288.3 | <0.01 |
| Independent | 1 | ARD | 8 | 2817.1 | <0.01 | 2817.1 | <0.01 | 2772.2 | <0.01 | 2772.2 | <0.01 | 1285.1 | <0.01 | 1285.1 | <0.01 |
| Independent | 2 | ER | 8 | 2684.4 | <0.01 | 2684.3 | <0.01 | 2619.5 | <0.01 | 2619.6 | <0.01 | 1234.6 | <0.01 | 1234.6 | <0.01 |
| Independent | 2 | SYM | 12 | 2651.7 | <0.01 | 2651.7 | <0.01 | 2586.9 | <0.01 | 2586.9 | <0.01 | 1217.9 | 0.01 | 1217.9 | 0.01 |
| Independent | 2 | ARD | 20 | 2625.9 | 0.04 | 2624.9 | 0.01 | 2589.3 | <0.01 | 2568.3 | <0.01 | 1215.1 | 0.03 | 1215.1 | 0.02 |
| Independent | 3 | ER | 18 | 2682.5 | <0.01 | 2682.4 | <0.01 | 2618.9 | <0.01 | 2618.9 | <0.01 | 1236.0 | <0.01 | 1234.8 | <0.01 |
| Independent | 3 | SYM | 24 | 2649.3 | <0.01 | 2655.2 | <0.01 | 2585.8 | <0.01 | 2585.8 | <0.01 | 1224.4 | <0.01 | 1225.6 | <0.01 |
| Independent | 3 | ARD | 36 | 2630.2 | <0.01 | 2630.3 | <0.01 | 2562.8 | <0.01 | 2564.9 | <0.01 | 1233.7 | <0.01 | 1233.8 | <0.01 |
| Dependent | 1 | ER | 1 | 4038.5 | <0.01 | 4038.4 | <0.01 | 3833.5 | <0.01 | 3833.5 | <0.01 | 1845.2 | <0.01 | 1845.2 | <0.01 |
| Dependent | 1 | SYM | 9 | 2764.0 | <0.01 | 2764.0 | <0.01 | 2753.0 | <0.01 | 2753.0 | <0.01 | 1258.8 | <0.01 | 1258.8 | <0.01 |
| Dependent | 1 | ARD | 18 | 2706.5 | <0.01 | 2706.5 | <0.01 | 2675.8 | <0.01 | 2696.7 | <0.01 | 1254.9 | <0.01 | 1254.9 | <0.01 |
| Dependent | 2 | ER | 4 | 3894.8 | <0.01 | 3894.8 | <0.01 | 3701.0 | <0.01 | 3700.9 | <0.01 | 1783.7 | <0.01 | 1783.6 | <0.01 |
| **Dependent** | **2** | **SYM** | **20** | 2659.4 | <0.01 | 2659.4 | <0.01 | 2596.0 | <0.01 | 2596.0 | <0.01 | **1207.8** | **0.97** | **1207.8** | **0.97** |
| **Dependent** | **2** | **ARD** | **38** | **2619.4** | **0.96** | **2616.3** | **0.99** | **2544.0** | **1.00** | **2553.6** | **1.00** | 1227.2 | <0.01 | 1224.3 | <0.01 |
| Dependent | 3 | ER | 9 | 3878.8 | <0.01 | 3878.8 | <0.01 | 3688.0 | <0.01 | 3688.0 | <0.01 | 1778.6 | <0.01 | 1778.5 | <0.01 |
| Dependent | 3 | SYM | 33 | 2636.7 | <0.01 | 2643.8 | <0.01 | 2588.2 | <0.01 | 2592.6 | <0.01 | 1222.2 | <0.01 | 1222.2 | <0.01 |
| Dependent | 3 | ARD | 60 | 2632.3 | <0.01 | 2635.5 | <0.01 | 2561.7 | <0.01 | 2580.9 | <0.01 | 1259.9 | <0.01 | 1266.0 | <0.01 |

The k rate is the number of evolutionary transition rate parameters independently estimated by a given model. The AICc is the sample size-corrected Akaike information criterion, while the AICcWt is the weighted AICc that provides the relative likelihood of each model per dataset version. The text in bold corresponds to the best-fitted model for each dataset version according to the AICc. The assembly of datasets v1 to v6 is summarized in Table 3 and described in the Methods. ER all rates are equal, SYM rates between any two states do not differ, ARD all rates differ.

**Table 2 Ranking of rates of evolutionary transitions from a combined mycorrhizal strategy and drought adaptation state to another.**

| Hidden rate | Mycorrhizal strategy | Drought adaptation | | Mycorrhizal strategy | Drought adaptation | Evolutionary rate (n° of transitions/Ma) | | |
|---|---|---|---|---|---|---|---|---|
| | | | | | | Best estimate | Lower CI | Upper CI |
| **R1** | **EEM** | **sensitive** | → | **EEM** | **tolerant** | **86,582511999017** | **65,450418567041** | **125,596365081402** |
| **R1** | **EEM** | **tolerant** | → | **EEM** | **sensitive** | **73,441795591613** | **52,327915630090** | **104,376960693156** |
| **R2** | **EEM** | **tolerant** | → | **EEM** | **sensitive** | **37,361861265910** | **31,894179724926** | **39,492300910717** |
| **R2** | **EEM** | **sensitive** | → | **EEM** | **tolerant** | **11,643466238308** | **11,128525164214** | **13,812425470750** |
| **R1** | **AM** | **sensitive** | → | **AM** | **tolerant** | **8,615462815502** | **7,843546713499** | **22,599962377315** |
| **R1** | **AM** | **tolerant** | → | **AM** | **sensitive** | **5,313692566879** | **4,705202251714** | **18,612166140747** |
| **R1** | **NR** | **tolerant** | → | **NR** | **sensitive** | **0,576387138006** | **0,522971570246** | **0,793727297506** |
| **R1** | **NR** | **sensitive** | → | **NR** | **tolerant** | **0,107610239498** | **0,088846456873** | **0,146718474560** |
| **R2** | **AM** | **tolerant** | → | **AM** | **sensitive** | **0,028298668362** | **0,025023426996** | **0,034710909449** |
| **R2** | **AM** | **sensitive** | → | **AM** | **tolerant** | **0,022721990898** | **0,019970548047** | **0,025909737396** |
| **R2** | **NR** | **sensitive** | → | **NR** | **tolerant** | **0,014913077227** | **0,012030149387** | **0,020858134381** |
| R2 | AM | sensitive | → | NR | sensitive | 0,007103600272 | 0,006072843735 | 0,009334948026 |
| R2 | NR | tolerant | → | AM | tolerant | 0,006625326669 | 0,005333006628 | 0,007312956799 |
| R2 | NR | sensitive | → | AM | sensitive | 0,005742399929 | 0,005173842022 | 0,008469336549 |
| R2 | EEM | tolerant | → | NR | tolerant | 0,001931483609 | 0,001585817919 | 0,002662471025 |
| R2 | AM | tolerant | → | NR | tolerant | 0,001382519339 | 0,001049122879 | 0,001879897364 |
| R2 | AM | tolerant | → | EEM | tolerant | 0,000913520672 | 0,000500702598 | 0,001223465552 |
| R1 | AM | sensitive | → | NR | sensitive | 0,000810931106 | 0,000410652141 | 0,001036289456 |
| **R2** | **NR** | **tolerant** | → | **NR** | **sensitive** | **0,000426436243** | **0,000000000598** | **0,001265007536** |
| R1 | EEM | sensitive | → | AM | sensitive | 0,000401966831 | 0,000256304564 | 0,000834565970 |
| R2 | NR | sensitive | → | EEM | sensitive | 0,000398557871 | 0,000220824529 | 0,000579460185 |
| R1 | AM | tolerant | → | EEM | tolerant | 0,000393451378 | 0,000262818503 | 0,000612884496 |
| R1 | NR | sensitive | → | AM | sensitive | 0,000375745460 | 0,000022671739 | 0,000620599513 |
| R2 | EEM | tolerant | → | AM | tolerant | 0,000311915653 | 0,000000000776 | 0,000497703546 |
| R1 | EEM | tolerant | → | AM | tolerant | 0,000283991264 | 0,000077594671 | 0,000487516580 |
| R2 | NR | tolerant | → | EEM | tolerant | 0,000237882397 | 0,000216588346 | 0,000371126212 |
| R1 | NR | sensitive | → | EEM | sensitive | 0,000214332755 | 0,000126427676 | 0,000337498115 |
| R1 | AM | tolerant | → | NR | tolerant | 0,000164730021 | 0,000154474989 | 0,000240585314 |
| R1 | AM | sensitive | → | EEM | sensitive | 0,000150354607 | 0,000047872599 | 0,000381895635 |
| R2 | EEM | sensitive | → | AM | sensitive | 0,000119255628 | 0,000075964367 | 0,001083389642 |
| R1 | EEM | sensitive | → | NR | sensitive | 0,000000001000 | 0,000000000661 | 0,000277361068 |
| R1 | NR | tolerant | → | AM | tolerant | 0,000000001000 | 0,000000000456 | 0,000120639383 |
| R1 | NR | tolerant | → | EEM | tolerant | 0,000000001000 | 0,000000000470 | 0,000199421575 |
| R2 | AM | sensitive | → | EEM | sensitive | 0,000000001000 | 0,000000000376 | 0,000191247207 |
| R2 | EEM | sensitive | → | NR | sensitive | 0,000000001000 | 0,000000000461 | 0,000130162555 |
| R1 | EEM | tolerant | → | NR | tolerant | 0,000000000996 | 0,000000000511 | 0,000047209341 |

The hidden rate corresponds to the level (R1 or R2) of the hidden phylogenetic factor influencing the evolution of the observed characters' mycorrhizal strategy and drought adaptation (Methods). Rows with text in bold inside the table correspond to evolutionary transitions between drought adaptation states within a given mycorrhizal strategy state, while rows with regular text inside the table correspond to transitions among mycorrhizal strategy states within a given drought adaptation state. The values are the average of the best rate estimates and the respective lower and upper 95% confidence interval (CI) for each estimate provided by the models best fitted to dataset v1 to v6 (see also Supplementary Table 1). For details on the assembly of dataset versions, see Table 3 and Methods. The sample size of extant plant states in each dataset version is provided in Table 4.
AM arbuscular mycorrhizal state, EEM combined ectomycorrhizal and ericoid mycorrhizal state, NR naked root state, i.e., non-mycorrhizal alone (dataset v5 and v6) or together with facultatively AM (dataset v1 to v4).

---

**Table 3 Summary of the assembly of the dataset versions analyzed.**

| Dataset | Drought adaptation | Mycorrhizal strategy | Phylogenetic tree |
|---|---|---|---|
| v1 | Prioritizing DatasetID 92 | Based on FungalRoot | Phylogeny mapped against GBMB |
| v2 | Prioritizing DatasetID 92 | Based on FungalRoot | Phylogeny mapped against GBOTB |
| v3 | Prioritizing DatasetID 49 and 98 | Based on FungalRoot | Phylogeny mapped against GBMB |
| v4 | Prioritizing DatasetID 49 and 98 | Based on FungalRoot | Phylogeny mapped against GBOTB |
| v5 | Prioritizing DatasetID 92 | Based on Bueno dataset | Phylogeny mapped against GBMB |
| v6 | Prioritizing DatasetID 92 | Based on Bueno dataset | Phylogeny mapped against GBOTB |

DatasetID refers to the ID number of the original dataset provided by the TRY—Plant Trait Database[27]. FungalRoot refers to the dataset published by ref. [26], while the Bueno dataset refers to the dataset published by ref. [25]. GBMB and GBOTB refers to plant phylogeny versions published by ref. [24]. For a description of the dataset version assembly, see Methods.

---

**Mycorrhizal symbiosis accelerates the evolutionary process of plant adaptation to drought**. The fastest transition rates over the past 325 My of plant evolution occurred primarily between the drought adaptation states within a given mycorrhizal strategy state, regardless of the hidden rate category (Table 2); the only exception to this was the rate of transitions from the drought tolerant to the sensitive state in lineages with an NR state in the hidden rate category two, which occurred at a relatively slow rate.

**Table 4 Number of plant species grouped by their drought adaptation and mycorrhizal strategy state according to each dataset version analyzed.**

| Drought adaptation | Mycorrhizal strategy | Dataset | | |
|---|---|---|---|---|
| | | v1 or v2 | v3 or v4 | v5 or v6 |
| Sensitive | AM | 408 | 390 | 232 |
| | EEM | 169 | 157 | 83 |
| | NR | 181 | 182 | 110 |
| Tolerant | AM | 637 | 655 | 238 |
| | EEM | 177 | 189 | 104 |
| | NR | 66 | 65 | 20 |
| Total | | 1638 | 1638 | 787 |

The assembly of datasets v1 to v6 is summarized in Table 3 and described in the Methods. *AM* arbuscular mycorrhizal state, *EEM* combined ectomycorrhizal and ericoid mycorrhizal state, *NR* naked root state, i.e., non-mycorrhizal alone (dataset v5 and v6) or together with facultatively AM plants (dataset v1 to v4).

**Table 5 Phylogenetic imbalance ratio per dataset version.**

| Dataset | NIR | CI | PIR |
|---|---|---|---|
| v1 | 0.34860 | 0.00636 | 0.00222 |
| v2 | 0.34860 | 0.00636 | 0.00222 |
| v3 | 0.36020 | 0.00676 | 0.00243 |
| v4 | 0.36020 | 0.00676 | 0.00243 |
| v5 | 0.27700 | 0.01415 | 0.00392 |
| v6 | 0.27700 | 0.01415 | 0.00392 |

The phylogenetic imbalance ratio (PIR) is the product of the normalized imbalance ratio (NIR) and consistency index (CI) calculated as described in the Methods. The assembly of datasets v1 to v6 is summarized in Table 3 and described in the Methods. The maximum PIR threshold is 0.1, as recommended by ref. [34].

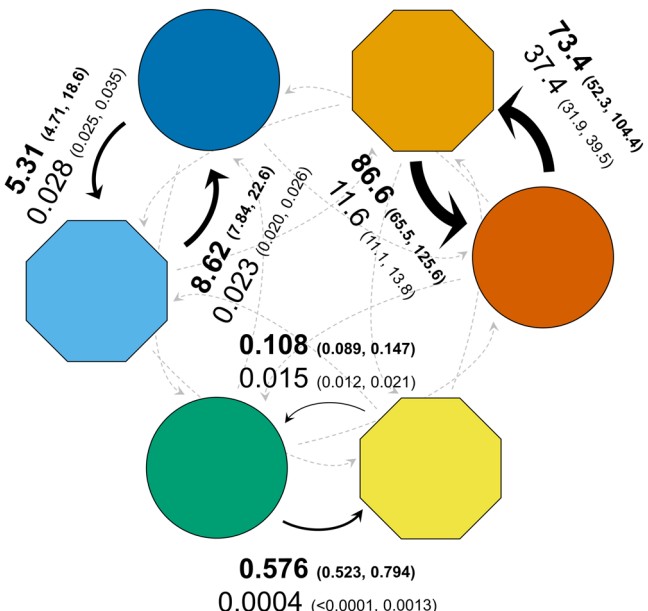

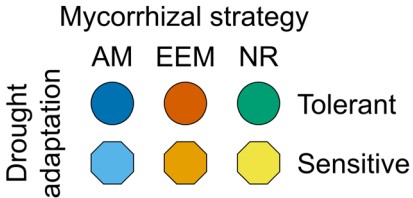

**Fig. 2 Transition rates of the dependent evolution for plant mycorrhizal strategy and drought adaptation.** Values are averages of the best rate estimates (per mega-annum; Ma) for the evolutionary transitions between a plant mycorrhizal strategy and drought adaptation state to another based on the best rate estimates obtained from the individual models best fitted to the dataset v1 to v6. The 95% confidence intervals for the best rate estimates are shown between brackets. Values provided in bold and regular font correspond to rates in the hidden rate category one (R1) and two (R2), respectively. The width of the arrows is proportional to rates in R1. A dotted gray arrow indicates that the rate is lower than 0.01 transitions per Ma. The average values of the best estimations for all the evolutionary transition rates and the respective 95% confidence intervals are shown in Table 2. The best rate estimates and the 95% confidence intervals for each dataset version are shown in Fig. 3 and Supplementary Table 1. For details on the assembly of dataset versions, see Table 3 and Methods. The sample size of extant plant states in each dataset version is provided in Table 4.

This indicates that drought adaptation in land plants mainly occurs under the influence of relatively rapid evolutionary processes, compared to that of the more conservative evolution of the mycorrhizal strategy. Among the fastest rates of evolutionary transitions between drought adaptation states, the top four occurred in lineages with an EEM state, regardless of the hidden rate category (Fig. 2 and Table 2). These transition rates were, on average—among all transition directions and hidden rate categories—~15 and 300 times faster than in lineages with the AM and NR state, respectively (Fig. 2). This conclusion was largely robust in relation to uncertainties in the data and in the rate parameter estimates, with only a few exceptions (Fig. 3). For the estimated rates of transitions between drought adaptation states in the hidden rate category one, only dataset v5 showed weak differences between the EEM and AM lineages (Fig. 3a, b), while in the hidden rate category two, only the dataset v4 to v6 showed weak differences between the AM and NR state (Fig. 3c, d). All the other differences among estimated rates of transitions between drought adaptation states within the different mycorrhizal strategy states had robust estimates across all six dataset versions analyzed (Fig. 3). Furthermore, the differences between the rates of gains and losses of drought tolerance depended on both the mycorrhizal strategy and hidden rate category. In the EEM lineages, which included many globally distributed plants of the Fagales, Malpighiales, Pinales (Gymnosperms), and Ericales, among others (Fig. 1b), the rate of losses of drought tolerance was significantly faster than the rate of gains in the hidden rate category two, while in the hidden rate category one the rate of gains was not significantly different from the rate of losses (Fig. 2). In the AM lineages, the rates of gains and losses of drought tolerance were not significantly different from each

other, regardless of the hidden rate category (Fig. 2). The AM lineages included many plants of the Rosales, Poales (monocots), Asterales, Fabales, Lamiales, among many other orders (Fig. 1b). In lineages with the NR strategy, which included many plants of the Poales, Caryophyllales, Alismatales (monocots), Brassicales, and Lamiales, among others (Fig. 1b), the rate of losses of drought tolerance was significantly faster than the rate of gains in both hidden rate categories (Fig. 2).

In terms of evolutionary shifts in mycorrhizal strategy states, the fastest rate observed was the losses of AM symbiosis— i.e., evolutionary transitions from the AM to the NR state—in drought-sensitive lineages, either within the hidden rate category one or two (Table 2). However, the differences between these and other estimated rates were weak (Table 2). In addition, other differences among rate estimates for mycorrhizal strategy transitions within each drought adaptation state were weak,

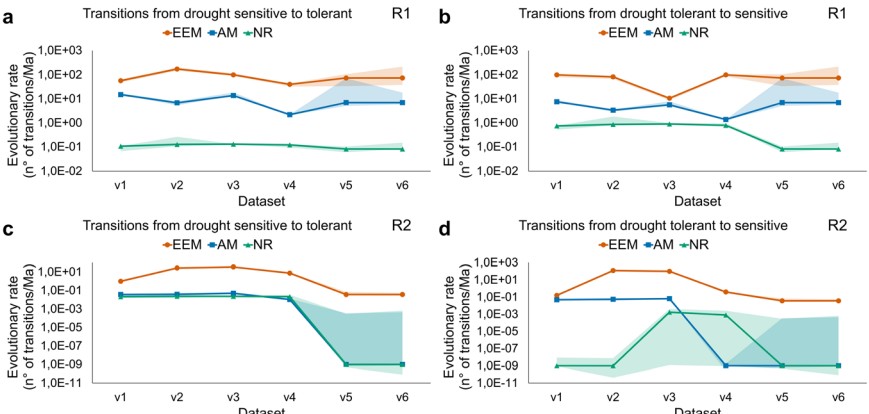

**Fig. 3 Sensitivity analysis on the rates of evolutionary transitions between drought adaptation states within a given mycorrhizal strategy state.** The variation of the best rate estimates in logarithmic scale is shown, as determined by the models best fitted to dataset v1 to v6, for the evolutionary transitions from drought-sensitive to tolerant (**a**) and drought tolerant to sensitive (**b**) in the hidden rate category one (R1), and from drought-sensitive to tolerant (**c**) and drought tolerant to sensitive (**d**) in the hidden rate category two (R2), as influenced by the arbuscular mycorrhizal (AM) state, the combined ectomycorrhizal and ericoid mycorrhizal (EEM) state, or the naked root (NR) state, i.e., non-mycorrhizal alone (dataset v5 and v6) or together with the facultatively AM plants (dataset v1 to v4). The assembly of datasets v1 to v6 is summarized in Table 3 and described in the Methods. Shaded areas around the transition rates represent the 95% confidence intervals for the best rate estimates. The sample size of extant plant states in each dataset version is provided in Table 4.

sensitive to data uncertainty, and dependent on the hidden rate category (Table 2 and Supplementary Table 1).

Overall, my results provide evidence that lineages forming consistent mycorrhizal symbiosis accelerate the evolutionary processes of gains and losses of plant tolerance to drought, supporting my second hypothesis. Moreover, these results constitute a quantitative demonstration of a long-term evolutionary advantage of host-microbe associations, and indicate that one of the main evolutionary advantage of mycorrhizas for land plants is related to rapid evolutionary changes in drought adaptation, rather than gains of tolerance alone.

## Discussion

With a looming climate change leading to the increasing manifestation of droughts across the globe, it is imperative to advance our understanding of how drought adaptation evolves in plants. My study provides a significant contribution by showing that the evolution of plant adaptation to drought and mycorrhizal strategy depended not only on one another but also on a hidden phylogenetic factor. It remains to be determined whether this factor is linked to root traits dependent or independent from the mycorrhizal strategy[37–39], or instead to other, yet under-examined, plant traits. Moreover, my study shows that, regardless of the hidden factor, a consistent mycorrhizal symbiosis can play a key facilitator role in the evolutionary processes of both gains and losses of drought tolerance in land plants. This constitutes a quantitative demonstration that host-microbe associations can contribute to shape the evolution of host adaptive traits essential to cope with critical environmental variation across temporal and spatial scales.

Among ecological factors interacting with drought to influence vegetation shifts or stability, the most well-studied are pests, pathogens, and grazing, while the role of beneficial biotic interactions are often neglected[11], including mycorrhizas. My study, although based on an evolutionary time scale too coarse to be fully informative at finer ecological timescales, when combined with previous eco-physiological studies[7,8], supports the consideration of mycorrhizal strategy when studying the impact of droughts on vegetation to help us fully appreciate the mechanisms underpinning temporal variation and trajectory of

responses, particularly when considering larger timescales of post-drought monitoring[11,14,40].

Mycorrhizas are well known to influence the physiological capacity of plants to tolerate drought, with many studies documenting direct and indirect mechanisms, such as mycelial transport of water and nutrients, leading to beneficial effects on plant performance, including improved survival under drought[7,8,18–20]. Although less often reported, mycorrhizas can also have neutral or negative effects on plant water relations during drought[8]. My results show that mycorrhizas can promote the evolutionary gains of plant tolerance to drought nearly as much as the gains of sensitivity, which challenges a predominant perception of mycorrhizas primarily as enablers of drought tolerance in land plants[18–20], and suggests that functional diversity might be more important in this context than previously recognized. Future studies combining my results with empirical testing could help us better understand how ectomycorrhizas, ericoid mycorrhizas, and arbuscular mycorrhizas influence the host survival and fitness in selective environments where drought sensitivity is an eco-physiological advantage for the plants.

One pressing question arising from my results is: what factors remain sufficiently stable at an evolutionary time scale to allow mycorrhizal plant lineages to change more swiftly in their adaptation to drought? These factors do not seem to be directly linked with the abundance of plant genes in the toolkit that many lineages carry across generations to form symbioses[41,42]. This is because ectomycorrhizal, ericoid mycorrhizal, and non-mycorrhizal lineages have higher degrees of symbiosis gene deletions compared with that of AM lineages[41,42], while my results show that changes in drought adaptations are faster in the ectomycorrhizal, ericoid mycorrhizal, and AM lineages, compared to that of the non-mycorrhizal and facultatively AM lineages. Furthermore, symbiosis-related molecular pathways harbored in both mycorrhizal and non-mycorrhizal plants, such as strigolactones and phenylpropanoids[43–45], are unlikely to play a role on their own. However, these molecular pathways may contribute indirectly by allowing plants to control their mycorrhizal strategy, and in this way, provide access to the factors that could contribute more directly to shape the evolution of plant adaptation to drought. A more plausible direct mechanism might be the imperfect vertical transmission of fidelity[46]. Although

mycorrhizal fungi are not vertically transmitted, as are other microbes in other host systems[46], and host plants have to reacquire fungi from the soil every generation[47], the mycorrhizal strategy is vertically heritable[22,41,42]. Moreover, it seems safe to assume that mycorrhizal fungal guilds are relatively stable entities at an evolutionary time scale based on observations of fungal fossils, chemistries, and genomes[48–51]. The transgenerational transmission of partner fidelity in the ectomycorrhizal and ericoid mycorrhizal plant lineages seems to be relatively low, with about 6000 ectomycorrhizal host species associating with more than 20,000 ectomycorrhizal fungal species, and some of these fungi can form ericoid mycorrhizas with many plants[52,53]. In contrast, the AM plant lineages seem to be relatively faithful, with more than 200,000 AM host species associating with only about 300 AM fungal species[52,54]. Functional variation of mycorrhizal effects on plant water relations during droughts has been documented, and are particularly recognized for ectomycorrhizas, where mycelium exploration types vary from close contact explorations, with smooth mycorrhizal tips having only a few short emanating hyphae, to extremely long explorative distances with highly differentiated rhizomorphs[7,8,55]. Hence, a low host fidelity across generations might enhance offspring phenotypic variance due to the functional variation[46] among mycorrhizal fungi. Under a fluctuating natural selection, due to critical environmental changes in precipitation over time and land surface, an enhanced offspring phenotypic variance is likely to ensure that at least some plant individuals are able to maintain a non-zero fitness in any given time step, lowering the likelihood of extinction, and increasing the rate of adaptation[46]. Hence, differences in host plant fidelity towards the fungal partners could potentially help to explain how the contrasting mycorrhizal strategies differently shape the speed of evolution of drought adaptation in land plants.

In sum, my study shows that the evolution of plant adaptation to critical environmental change in water availability across global climates is inherently dependent on mycorrhizas. The mycorrhiza-mediated drought adaptation could be potentially linked to the host plant's fidelity toward the fungal partners. Future investigations of this and related working hypotheses under past and future climate change scenarios are needed to yield further insights into the role of mycorrhizas in plant evolution.

## Methods

**Data on plant adaptation to drought.** I downloaded trait data on 'species tolerance to drought' (TraitID 30) for 3324 plant species from the Try—Plant Trait Database[27], distributed among seven original datasets. This trait was renamed here as 'drought adaptation' for a clear distinction between trait name and state. From the original datasets obtained, only the larger four (DatasetID 49, 68, 92, and 98) showed significant species overlaps with the FungalRoot dataset (the larger reference used here to assign the mycorrhizal strategy states, as described below). Because DatasetID 49 and 68 were remarkably identical, with the former being slightly larger and accompanied by detailed information on data determination, only the three original datasets DatasetID 49, 92, and 98 were used in this study[56–58]. DatasetID 49 is the Tree Tolerance Database and was produced by the Estonian University of Life Sciences (Estonia), DatasetID 92 is the PLANTSdata database and was produced by the United States Department of Agriculture (USA), and DatasetID 98 is the New South Wales Plant Traits Database and was produced by the Macquarie University (Australia). The latter covers areas of the Australian continent that are not covered by the former two, and it was included to incorporate plants living in arid and xeric environments of this part of the Globe.

To standardize data interpretation across classification systems of the different original dataset references, and to streamline model estimation of parameters, the drought adaptation was simplified into a binary variable, with species having either a "tolerant" or "sensitive" state, following a previous approach[17]. The underlying rationale for this standardization is based on the notion that all classification systems were built with the same purpose of comparing drought tolerance among plants in their biomes, and that each system can be equally split into two levels. This standardization was needed to achieve a global coverage that incorporates plants from different environments. For DatasetID 49, the classification system

varied continuously between 0 and 5, with 1, 2, 3, 4, and 5 standing for 'very intolerant', "intolerant", "moderately tolerant", "tolerant", and "very tolerant" to drought, respectively[56]. Therefore, species with an original value of less than 2.5 were assigned here with a sensitive state, whereas the remaining species were assigned with a tolerant state. For DatasetID 92, the classification system varied categorically among "none", "low", "medium", and "high" tolerance to drought. Hence, the species with the original value "none" or "low" tolerance were assigned here with a sensitive state, whereas the species with the original value "medium" or "high" tolerance were assigned with a tolerant state. For DatasetID 98, the classification system varied categorically among "none, dies off in dry conditions", "medium, dies off after several months", "fairly drought resistant", and "very drought resistant". Thus, the species with the original value "none, dies off in dry conditions" or "medium, dies off after several months" were assigned here with a sensitive state, whereas the species with original value "fairly drought resistant" or "very drought resistant" were assigned with a tolerant state. These assignments are considered here to provide suitable equivalences among classification systems based on the underlying rationale mentioned above.

Although this approach allowed state standardization across datasets, it still led to assignment mismatches for some species, unveiling data uncertainty (addressed below in sensitivity analysis). This occurred only for the assignments using DatasetID 92 against that of 49 or of 98. However, by considering these mismatches, I assembled dataset versions with partial differences in plant adaptation to drought by prioritizing the assignments based either on DatasetID 92 or on both DatasetID 49 and 98 together (described below in the assembly of dataset versions).

**Data on plant mycorrhizal strategy.** Data on plant mycorrhizal types were obtained from two sources, the FungalRoot database[26] and an alternative dataset[25] (hereafter the Bueno dataset). The FungalRoot is the largest database of its kind ever assembled, containing mycorrhizal assignments for 14,347 plant genera, and has been successfully employed in previous large-scale studies[38,59]. The main mycorrhizal types included in the FungalRoot are the arbuscular mycorrhizal (AM), facultatively AM, ectomycorrhizal, dual ectomycorrhizal and AM, ericoid mycorrhizal, orchid mycorrhizal, and non-mycorrhizal type, which follows the mycorrhiza definitions previously proposed[5,60]. However, because these definitions are still, in part, a matter of debate[25,61–64], particularly around the assignment of the facultatively mycorrhizal plants, to address the issue of data uncertainty, I generated partially different dataset versions of mycorrhizal strategy using the relatively smaller Bueno dataset (described below in assembly of dataset versions). The Bueno dataset includes assignments of AM, ectomycorrhizal, ericoid mycorrhizal, and non-mycorrhizal types for a total of 1362 plant species.

Similar to the data on drought adaptation, to streamline model estimation of parameters, the mycorrhizal strategy was simplified here into a ternary variable following a previous approach[29]. Central to this approach is the observation that plants have evolved different strategies for investing photosynthetically fixed carbon to compete for limiting soil resources: (1) by scavenging for plant-available nutrients mainly in symbiosis with AM fungi[29,30] (AM state); (2) by mining organic-bound nutrients primarily in symbiosis with ectomycorrhizal or ericoid mycorrhizal fungi[29,31,32] (EEM state); or (3) by taking up resources mostly via the absorptive surface of their own naked roots[5,22,29] (NR state). In addition, monophyletic mycorrhizas were not considered here alone to minimize phylogenetic comparative issues related to single evolutionary transitions[34]. Thus, to determine the mycorrhizal strategy using the FungalRoot database, plant species belonging to genera with AM type were assigned here with an AM state, while those belonging to genera with either ectomycorrhizal or ericoid mycorrhizal type were assigned with an EEM state. In addition, because many dual mycorrhizal plants tend to be dominated by AM only during their seedling stage[5,65], the plant species belonging to genera with dual ectomycorrhizal and AM type were considered here to have a predominant EEM state. Furthermore, the naked root state is considered here as a negative control for obligate mycorrhizal symbiosis. Therefore, because many non-mycorrhizal and facultatively AM plants are habitat or nutritional specialists where mycorrhizas are described as less relevant[5,22], the plant species belonging to genera with either of these two types were assigned here with a predominant NR state. For the facultatively AM plants in particular, these have relatively lower frequencies of mycorrhizal association[26], and as a lowered frequency of mycorrhizal association is often considered an acceptable form of negative control to characterize the functions of mycorrhizas in the field[66], this further justifies their inclusion in the predominantly naked root state. Moreover, the detected three plant species that belong to genera with the orchid mycorrhizal type were excluded from the analysis. To determine the mycorrhizal strategy using the Bueno dataset, the plant species with the AM, ectomycorrhizal or ericoid mycorrhizal, and non-mycorrhizal type were assigned here with the AM, EEM, and NR state, respectively.

**Plant phylogeny data.** I obtained two plant phylogeny versions (GBMB and GBOTB) from ref. [24], which is the most broadly inclusive, time-calibrated plant phylogeny construction published to date. This phylogeny has been successfully employed in previous large-scale studies[15,27]. Both phylogeny versions were constructed with GenBank taxa. However, GBMB has 79,874 taxa and backbone provided by ref. [67], while GBOTB has 79,881 taxa and a backbone provided by the

Open Tree of Life version 9.1[24]. These two phylogeny versions were analyzed here to account for potential phylogenetic uncertainty. To assign the phylogenetic relationships among species, both phylogenetic trees were pruned by keeping only the tips whose labels matched the names of the species included in the assembled datasets (described below in the assembly of dataset versions).

**Assembly of dataset versions**. Based on the data assignments described above, a total of six partially different dataset versions were assembled, named here as dataset v1 to v6. A summary of the dataset assembly is provided in Table 3 and the files of the respective phylogenetic trees and data frames are provided in Supplementary Data 1 (found at https://doi.org/10.5061/dryad.3ffbg79nx). The dataset v1 to v4 had each 1638 species, while datasets v5 and v6 had each 787 species (Table 4). Of the total species analyzed, 22%, 9%, and 8% were always considered to be drought-sensitive and to have AM, EEM, and NR state, while 38%, 10%, and 3% were always considered to be drought tolerant and to have AM, EEM, and NR state, respectively (Supplementary Fig. 1). The remaining 9% of the species had uncertain drought adaptation and mycorrhizal strategy depending on the assembled dataset version (Supplementary Fig. 1). The six dataset versions were analyzed separately to determine the goodness of fit, and the resulting estimated rate parameters were used to generate averages and to characterize the impact of data uncertainty on main conclusions (described below in sensitivity analysis).

**Global mapping of species distribution**. I downloaded the georeferenced occurrences (latitude and longitude) for vascular plant species from the Global Biodiversity Information Facility[28], which included 457,547 occurrences distributed among 25,779 species. Then, to validate the global scale of my analysis, I used this georeferenced data to assign occurrences to the maximum number of species included in the assembled dataset versions (Supplementary Data 2 found at https://doi.org/10.5061/dryad.3ffbg79nx).

**Modeling of plant character evolution**. To test for correlated character evolution and estimate evolutionary transition rates between states, I ran 18 HMMs of incrementing complexity, using the corHMM function of the R package corHMM v2.1[33]. The HMMs are important centerpieces to understand character evolution[33,34,68], and have been previously employed with great efficiency, rigor, and objectivity in large-scale studies[69–71]. Briefly, the corHMM function takes a phylogenetic tree and state data to estimate among several parameters the transition rates among states of discrete characters. The models tested here included either a dependent or independent mode of evolution for the characters "drought adaptation" and "mycorrhizal strategy". Moreover, the corHMM function allows to detect the occurrence of unobserved (or hidden) phylogenetic factors that have either promoted or constrained the evolutionary processes of the observed characters—including the influence of heterogenous diversification along phylogenies—controlling for phylogenetic bias[33,35,36,68]. Thus, each model tested here included either one, two, or three categories of hidden rates. Moreover, each model included a structure of either homogeneous evolution (all evolutionary transition rates are equal; ER), partially heterogenous evolution (only evolutionary transition rates between any two character states do not differ; SYM), or heterogenous evolution (all evolutionary transition rates differ; ARD)[33]. Each model was run with five replicated starts, resulting in a total of 810 independent optimization exercises (output files provided in Supplementary Data 3 at https://doi.org/10.5061/dryad.3ffbg79nx). The corHMM function was run in R v3.6.2 using a supercomputer cluster. The codes used to run the HMMs on each dataset version are provided in Supplementary Note 1 at https://doi.org/10.5061/dryad.3ffbg79nx. Finally, I compared the sample size-corrected Akaike information criteria (AICc) among models to select the best-fitted one and to obtain the respective estimated evolutionary transition rates. These rates are based on estimated numbers of transitions that occurred over 325 million years of evolution.

**Sensitivity analysis**. I analyzed the robustness of model fitness and of estimated evolutionary transition rates in relation to three primary sources of data uncertainty: (1) adaptation to drought data; (2) mycorrhizal strategy data; and (3) phylogenetic tree backbone. To this end, each of the 18 HMMs (described above in modeling of plant character evolution) were run on each of the six dataset versions (Table 3), using five replicated starts for each model and dataset version combination. The robustness of model selection was evaluated by comparing how the different dataset versions changed the model ranking based on the AICc. To analyze the uncertainty in rate parameter estimates for the different dataset versions, and to obtain the respective 95% confidence intervals, each best-fitted model for each dataset version was run in the ComputeCI function of the R package corHMM v2.1[33], using 5000 sampled points. This function uses the R package dentist to sample points around a specified distance from the maximum likelihood estimates. The codes used to run the ComputeCI function on each best-fitted model and the respective output files are provided in Supplementary Note 1 and Supplementary Data 4 (found at https://doi.org/10.5061/dryad.3ffbg79nx), respectively. Finally, the robustness of the evolutionary transition rates in relation to data and estimate uncertainties was performed by visual comparison of how the

different dataset versions changed the best rate estimates and confidence intervals as inferred by the respective best-fitted models.

**Determining the phylogenetic imbalance ratio**. Phylogenetic comparative methods rely on whether the data contain sufficient information to inform inference[34,72]. Hence, to avoid erroneous detection of correlated evolution due to limited evolutionary sample size and/or state phylogenetic imbalance (e.g., due to a potential imbalance in phylogenetic diversification among mycorrhizal strategy and/or drought adaptation states), it is recommended that each dataset holds a phylogenetic imbalance ratio (PIR) below 0.1[34]. To evaluate this, I have calculated the PIR for each of the six dataset versions analyzed here following the formula recently proposed by Gardner & Organ:[34]

$$PIR = NIR \times CI \qquad (1)$$

In this formula (*i*), CI is the consistency index and was calculated using the CI function in the R package "phangorn", while *NIR* is the normalized imbalance ratio calculated using the following formula (*ii*):

$$NIR = \frac{T\max - T\min}{n}, \qquad (2)$$

where *n* is the size of the dataset and *T*max and *T*min is the maximum and minimum frequency of a state, respectively.

**Reporting summary**. Further information on research design is available in the Nature Portfolio Reporting Summary linked to this article.

## Data availability

The data generated and analyzed during the current study are provided as Supplementary Data and are available in the Dryad digital repository at https://doi.org/10.5061/dryad.3ffbg79nx.

## Code availability

The code used in this study is provided in Supplementary Note 1 and is available in the Dryad digital repository at https://doi.org/10.5061/dryad.3ffbg79nx.

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

## Acknowledgements

Marco Cosme was supported by the European Commission's grant H2020-MSCA-IF-2018 "SYMBIO-INC" (GA 838525)". Computational resources were provided by the supercomputing facilities of the Université catholique de Louvain (CISM/UCL) and the Consortium des Équipements de Calcul Intensif en Fédération Wallonie Bruxelles (CÉCI) funded by the Fond de la Recherche Scientifique de Belgique (F.R.S.-FNRS) under convention 2.5020.11 and by the Walloon Region. The author is thankful to Stéphane Declerck for office space and to James Boyko, Damien François, and Jacob Gardner for methodological clarifications.

## Author contributions

M.C. designed the research, collected and analyzed the data, interpreted the results, and wrote the manuscript.

## Competing interests

The author declares no competing interests.
