## [Peer Review File · Communications Biology]

nature portfolio

Peer Review FileReviewers' comments:

Reviewer #1 (Remarks to the Author):

Review COMMSBIO-13-4090-0

Mycorrhizas drive the evolution of plant adaptation to drought

General Review: This is an review study designed to evaluate the influence of mycorrhizal status on the adaptation of seed plants to drought. The study encompassed a total of 1362 plants representing angiosperms and gymnosperms lineages, which included the mycorrhizal associations and drought tolerance. The author hypothesized that a) there is a codependency between mycorrhizal fungi and plants in their evolutionary way to adapt to drier ecosystems and b) the speed of adaptation is influenced by the type of mycorrhizal association. They use phylogenetic tools, including evolutionary models, to test their hypotheses, and found compelling support for the idea of codependency between mycorrhizal fungi and host; and that rates of evolution in plant adaptation to drier conditions were influenced by their mycorrhizal partnership.

Specific comments:

This is an interesting article that explores an important aspect of the integration of symbiotic entities, such as the adaptation to climatic shifts. The paper explores a timely interesting aspect of mycorrhizal evolution little explored to date, representing an important step to understand the interactions between roots and soil microbial communities. Possibly, the most interesting result of this manuscript is the suspected evolutionary boost that fungal communities provide to plants when adapting to stressful conditions, promoting adaptation to stressful conditions. This could represent an important advance toward understanding other ecological processes such as soil respiration, carbon accrual and microbial metabolic activity in soils.

However, I think the paper is not suitable for publication at this stage. First, the paper lacks clarity on the objectives. The statement that the drought adaptation and mycorrhizal strategy depend on each other, does not define clearly the mechanisms that may explain this trend, neither exclude additional aspects that could be involved in the evolutionary changes. The author limited the analysis to estimate transitional rates between states, but the analysis lacked an explanation on what the co-dependency actually mean. In other words, what changes were present in plant roots and mycorrhizal fungi that may explain the coevolution. I believe that the authors missed the opportunity to test fully the idea of measuring the co-evolution rates because they did not fully tested the evolutionary correspondence between fungal and plant lineages as they adapted to drier conditions. Moreover, the author made strong statement about the definite necessity of mycorrhizal partners for adaptation to critical environmental changes (L 136-137) without exploring other alternative explanations that could influence the observed patterns. For instance, characteristic non-mycorrhizal plant lineages like Caryophyllales, Lamiales, Poales, and Brassicales, are also dominant groups in desert-like environments. But, they are also lineages with the highest speciation rates, mostly related to their adaptation to short lifespans, selfing sexual reproduction, and autochorous dispersion (Smith and Donoghue 2008). Thus, NM lineages may have a higher speciation rate not because they have abandoned the mutualism with mycorrhizal fungi but because they are largely short lived and herbaceous. Similar trends could be observed in wood adaptations of during the expansion of lineages, like Ericales and Fagales, to colder and drier new habitats (Zanne et al 2018). More generally, the findings in this manuscript may not be sensitive enough to exclude other aspects explaining plant diversification in novel environments.

Second, I had a hard time understanding the way the data set was standardized. The idea of equalizing different categorical classifications into a binary "sensitive" vs "tolerant" seems over simplified and potentially misleading. The classifications in the original studies included observation of plants in different biomes, with particular local adaptations to prevalent climatic conditions, and using different approaches in their drought tolerant groupings. Thus, what is consider "fairly drought resistant" in one study could be equivalent to "medium" in another with no objective way to equalize both. Similarly, the inclusion of facultative mycorrhizal and non mycorrhizal plants in the same "naked root" classification was poorly justified, particularly considering that the facultative stage can be meaningful

in the understanding of mycorrhizal transitions in plant evolution (see Maherali et al 2017 *American Naturalist*).

Finally, I believe the study lacks of insight with respect to recent literature, including important advances in root and fungal evolution. There are important developments in the evolution of root anatomy and morphology (Kong et al 2019 *Nat Comm*, Wang et al 2020 *Forest*, Valverde-Barrantes et al 2020 *New Phytologist*) that should be integrated in the paper, as part of the discussion. In addition, the manuscript is imbued with jargon and vague points. Statements like "The best fitted HMMs revealed that this correlated evolution was consistently influenced by an unobserved (hidden) phylogenetic factor with two state levels (Supporting Table S1), which has either promoted or constrained the speed of evolution of the observed characters drought adaptation and mycorrhizal strategy" (l 90-92) are too general to make a clear point about the conclusions in the paper. I recommend the author to be more specific in the analysis, targeting particular lineages and clarify how the transitions between drought tolerance stages influence mycorrhizal affiliation and evolution rates in both groups. Also, it will be important to look into the biogeography, and the ancestral reconstruction of the transitions, to have a better idea how these changes occurred in the past. As it is, I do not think the paper is ready for publication. I will be willing to make a more informed commenting on the paper, once this basic issues are addressed.

Reviewer #2 (Remarks to the Author):

Summary

This manuscript provides evidence for a macroevolutionary correlation between symbiotic type and drought resistance in seed plants. The author uses discrete character Markov models to test for evolutionary correlation between the two characters. They analyze several datasets with alternative character codings and phylogenetic trees to test the sensitivity of their results. They find evidence of dependent evolution and suggest that symbiotic type is highly influential in drought resistance evolution. For my part, I am primarily knowledgeable about the methodological approach used in this study and so will focus comments primarily in that area. Overall, I believe this manuscript makes a novel contribution to a growing literature of correlated evolution and there are few shortcomings in the methodological approach taken here.

Major Comments

The author does an admirable job in examining the sensitivity of their results to different datasets. However, I would like to see the uncertainty in the parameter estimates for these datasets and best fitting models. My reasoning is that the results of any correlated evolution analysis are dependent on the rate estimates. Although the differences between transitions in drought tolerance depending on the mycorrhizal association state are large, the variance in those estimates could be insightful for determining whether we should trust these seemingly large differences. I believe the most straightforward way to estimate a measure of parameter estimation uncertainty would be to conduct a parametric bootstrap for your best fitting models. This would mean simulating data under your best fit model several hundred times and refitting the model under its own simulated data. You would then on average expect the same parameters to be found as those simulated under, but there will be some amount of uncertainty in those estimates. That uncertainty/ variance could then be reported as standard error around the original estimates and give a general idea of the robustness of them.

Another useful addition could be the conducting of power analysis ala Boettiger et al. (2012). This would be a similar procedure as the parametric bootstrap, but with the explicit purpose of contrasting models against one another. I.e., does this dataset have the power to detect correlated evolution instead of independent evolution. I do not think that both of these analyses need to be considered, but at least some measure of power or uncertainty is necessary to be included. My preference would be to conduct a parametric bootstrap so that there is some measure of uncertainty around the parameter

estimates because the AIC indicates fairly strong evidence of the correlated model that power is unlikely to be an issue.

Minor Comments

Line 6: plant tolerate should be tolerance

Line 11: remove "that"

Line 141: "is markedly shaped by host-microbe associations". Although this is true given your results, the evidence of hidden rate categories suggest that host-microbe interactions are not the only factor influencing the evolution of drought tolerance. It is worth highlighting the uncertainty of this finding and that there are other mechanisms at play shaping drought tolerance.

References

Boettiger C., Coop G., Ralph P. 2012. Is Your Phylogeny Informative? Measuring the Power of Comparative Methods. *Evolution*. 66:2240–2251.

General Review: This is an review study designed to evaluate the influence of mycorrhizal status on the adaptation of seed plants to drought. The study encompassed a total of 1362 plants representing angiosperms and gymnosperms lineages, which included the mycorrhizal associations and drought tolerance.

Response: For clarification, this research paper encompasses 1,638 plants (lines 9, 54, 312, legends of Fig. 1 and Supporting Fig. S2, and Supporting Table S3), and thus incorporates more 276 plants than the number acknowledged by the reviewer.

The author hypothesized that a) there is a codependency between mycorrhizal fungi and plants in their evolutionary way to adapt to drier ecosystems and b) the speed of adaptation is influenced by the type of mycorrhizal association. They use phylogenetic tools, including evolutionary models, to test their hypotheses, and found compelling support for the idea of codependency between mycorrhizal fungi and host; and that rates of evolution in plant adaptation to drier conditions were influenced by their mycorrhizal partnership.

Response: I have hypothesized that: (1) the evolutions of drought adaptation and mycorrhizal strategy depend on each other; and (2) mycorrhizas markedly influence the speed of evolution of drought adaptation in land plants (lines 68-70). Drought adaptation is defined here as the adaptive changes from drought tolerance to drought sensitivity as well as from drought sensitivity to drought tolerance. Hence, it reflects plant adaption to drier habitats as much as plant adaption to wetter habitats in which drought sensitivity is an advantage for the plants. In response to a comment below that requires the manuscript to clarify the objective, this definition was added to the introduction (lines 71-74).

Specific comments:

This is an interesting article that explores an important aspect of the integration of symbiotic entities, such as the adaptation to climatic shifts. The paper explores a timely interesting aspect of mycorrhizal evolution little explored to date, representing an important step to understand the interactions between roots and soil microbial communities. Possibly, the most interesting result of this manuscript is the suspected evolutionary boost that fungal communities provide to plants when adapting to stressful conditions, promoting adaptation to stressful conditions. This could represent an important advance toward understanding other ecological processes such as soil respiration, carbon accrual and microbial metabolic activity in soils.

Response: I am thankful to the reviewer for the kind words of recognition of the interest, importance, and timely quality of the research documented in the manuscript, as well as for the suggested interdisciplinary implications for future research in ecology.

However, I think the paper is not suitable for publication at this stage. First, the paper lacks clarity on the objectives.

Response: The objective of the paper is to test the hypotheses that: (1) the evolutions of drought adaptation and mycorrhizal strategy depend on each other; and (2) mycorrhizas markedly influence the speed of evolution of drought adaptation in land plants (lines 68-70). This is now

made clearer in the text (lines 68). In addition, to clarify what these two hypotheses implicate, a further clarification on the definition of drought adaptation was added to the introduction (lines 71-74).

The statement that the drought adaptation and mycorrhizal strategy depend on each other, does not define clearly the mechanisms that may explain this trend, neither exclude additional aspects that could be involved in the evolutionary changes.

Response: I have difficulty to understand what kind of mechanisms the reviewer refers to, and how these can be studied using phylogenetic comparative methods. These methods allow us to study evolutionary processes along phylogenies of organisms (Beaulieu et al., 2013 Syst Biol; Werner et al., 2014 Nat Commun; Werner et al., 2018 Proc Natl Acad Sci USA; Boyko & Beaulieu, 2021 Methods Ecol Evol; Joly & Schoen, 2021 Curr Biol; Boyko & Beaulieu, 2022 Syst Biol), but to my knowledge, they are not adequate to unravel all underlying mechanisms driving evolution. Therefore, they cannot be used to define clearly the mechanisms that control all evolutionary trends, as these result from past biogeochemical and ecosystem processes that cannot be characterized using phylogenetic data only. Furthermore, I do not agree that the manuscript exclude additional aspects that could be involved in the evolutionary changes of the observed characters drought adaptation and mycorrhizal strategy. As stated in the results (now lines 91-95), and as recognized by Reviewer 2, the manuscript identifies the existence of an unobserved phylogenetic factor with two hidden rate categories that has either promoted or constrained the evolutionary process of the observed characters. This was done with as much clarity as that of previous studies (Werner et al., 2014 Nat Commun; Werner et al., 2018 Proc Natl Acad Sci USA; Joly & Schoen, 2021 Curr Biol).

The author limited the analysis to estimate transitional rates between states, but the analysis lacked an explanation on what the co-dependency actually mean.

Response: The dependent evolution among the two observed characters was explained, in my perspective, with the extent that the methods permit, i.e. it was explained that this correlated evolution signifies that, throughout the course of plant evolution, the rate of change in drought adaptation – i.e. evolutionary shifts between the drought sensitive and tolerant states – in a given lineage depends on the mycorrhizal strategy formed by that lineage – i.e. whether AM, EEM, or NR state –, and the rate of shifts among mycorrhizal states depends on the lineage's adaptation to drought (now in lines 82-86). In addition, it was explained in detail how the states of one character influence the rate of evolutionary transitions among the states of the other character, when robustly estimated (lines 113-130). To my understanding, this is already a robust explanation for the detected dependent evolution. I do not think that additional explanations could be made within the scope of the present manuscript.

In other words, what changes were present in plant roots and mycorrhizal fungi that may explain the coevolution. I believe that the authors missed the opportunity to test fully the idea of

measuring the co-evolution rates because they did not fully tested the evolutionary correspondence between fungal and plant lineages as they adapted to drier conditions.

Response: Although this is an interesting suggestion, I am afraid that it is unrealistic because it overlooks major state-of-the-art limitations in terms of data availability. The execution of this suggestion would require the availabilities of data on drought adaptation in multiple species of different mycorrhizal fungal guilds as well as of a broad time-calibrated phylogeny that includes the different mycorrhizal species. To my knowledge, both are not existing. The construction of a broad time-calibrated phylogeny alone involves an extensive effort that would merit its own publication, as it was the case for the phylogenies of seeds plants (Smith & Brown, 2018 *Am J Bot*), mammals (Álvarez-Carretero et al., 2022 *Nature*), and mushrooms (Varga et al., 2019 *Nat Ecol Evol*). Similarly, constructing a dataset on drought adaptation in different mycorrhizal fungal guilds is extremely challenging, and it would require methods that, to my knowledge, are not yet available. How can drought adaptation be measured across multiple species of different mycorrhizal fungi? In my five-year-long experience with monospecific cultivation of mycorrhizal fungi, I have not yet encountered a good solution for this question. Therefore, although interesting, the suggestion made by the reviewer is technically impossible at this stage. In regard to additional characters on plant roots, this creates two main issues. On one hand, including additional plant characters increases the complexity of the models, leading to greater parameterization with poorer model performance, as discussed by Boyko & Beaulieu (2021 *Methods Ecol Evol*). On the other hand, when I cross multiple plant traits, this leads to weaker datasets with reduced numbers of species because only a portion overlap (See consequences of dataset overlap in lines 213-321). The poorer model performance operating on weaker datasets yields weaker and less meaningful results and is therefore undesirable. Nonetheless, the discussion in my manuscript already offers a 300-word-long state-of-the-art explanation for how differences among mycorrhizal fungal guilds may contribute to explain their roles in shaping the speed of evolution of drought adaptation in plants (now lines 182-204), which is based on the recently proposed conceptual model of imprecise vertical transmission (Bruijning et al., 2022 *Nat Ecol Evol*). A sentence was added to the text to make this explanation clearer (lines 202-203). In addition, I now highlight and discuss in the text that other factors are at play in the correlated evolution (lines 91-100 and 142-149).

Moreover, the author made strong statement about the definite necessity of mycorrhizal partners for adaptation to critical environmental changes (L 136-137) without exploring other alternative explanations that could influence the observed patterns.

Response: In the former lines 136-137, I had stated that ‘My study provides a significant contribution by showing that the evolution of plant adaptation to drought and mycorrhizal strategy depended on one another.’ I do not think that this is a strong statement based on the provided evidence of correlated evolution. Moreover, alternative explanations due to phylogenetic bias are not considered a major issue when using Hidden Markov Models (Beaulieu et al., 2013 *Syst Biol*; Boyko & Beaulieu, 2021 *Methods Ecol Evol*), as these models consider the influence of unobserved phylogenetic factors, as previously stated in the methods (now lines 337-341) and now explained more clearly in both methods and results (lines 339-340 and 95-100).

For instance, characteristic non-mycorrhizal plant lineages like Caryophyllales, Lamiales, Poales, and Brassicales, are also dominant groups in desert-like environments. But, they are also lineages with the highest speciation rates, mostly related to their adaptation to short lifespans, selfing sexual reproduction, and autochorous dispersion (Smith and Donoghue 2008). Thus, NM lineages may have a higher speciation rate not because they have abandoned the mutualism with mycorrhizal fungi but because they are largely short lived and herbaceous. Similar trends could be observed in wood adaptations of during the expansion of lineages, like Ericales and Fagales, to colder and drier new habitats (Zanne et al 2018). More generally, the findings in this manuscript may not be sensitive enough to exclude other aspects explaining plant diversification in novel environments.

Response: Among the 247 species included in my dataset with a naked root strategy, 47 are non-mycorrhizal, drought tolerant, and belong to either the Caryophyllales, Lamiales, Poales, and Brassicales orders. My data shows that they represent a fraction of plants that have either weak or no mycorrhizal symbiosis, reflecting a known pattern in land plants (Brundrett 2009 Plant Soil; Wang & Qiu, 2006 Mycorrhiza). In contrast to what the reviewer seems to suggest, I could not find any evidence in Smith and Donoghue (2008 Science) indicating that non-mycorrhizal plant lineages have the highest speciation rates. In addition, I had difficulties to follow how the reviewer's rational relates to the manuscript's scope. For instance, explaining plant diversification in novel environments was not the aim of my manuscript. Instead, the aim of the manuscript was to test the hypotheses that: (1) the evolutions of drought adaptation and mycorrhizal strategy depend on each other; and (2) mycorrhizas markedly influence the speed of evolution of drought adaptation in land plants (now lines 68-70). What I could understand from the reviewer's comment is that the reviewer seems to have doubts on whether the chosen method can consider the influence of diversification. If this is the case, to test my hypotheses, I have used a well-established and widely accepted comparative phylogenetic method (Beaulieu et al., 2013 Syst Biol; Werner et al., 2014 Nat Commun; Werner et al., 2018 Proc Natl Acad Sci USA; Boyko & Beaulieu, 2021 Methods Ecol Evol; Joly & Schoen, 2021 Curr Biol; Boyko & Beaulieu, 2022 Syst Biol), which allows to detect the occurrence of unobserved (or hidden) phylogenetic factors that have either promoted or constrained the evolutionary processes of the observed characters, controlling for phylogenetic bias, as expressed in the methods (now lines 337-341). Although not explicitly expressed previously, when heterogenous phylogenetic diversification along the phylogeny has an important influence on the evolution of the observed characters, this influence is accounted by (and detected with) the tested models that analyze the phylogeny while considering site-specific rate heterogeneities. This is now expressed more clearly in the text (lines 95-97 and 339-340). In addition, as shown in the Supporting Table S1, for each of the six dataset versions analyzed, the model with two hidden rate categories was better fitted than the three-rate-category equivalent. This indicates that, although the rate heterogeneities matter, these were better grouped in a simple two-class organization. This information is now added to the text (lines 98-100). Finally, if there was worrying bias due to imbalanced phylogenetic diversification among observed character states (for instance, due strong contrasts in phylogenetic diversification among lineages with different mycorrhizal strategies), such diversification bias would have been picked up by the phylogenetic imbalance ratio, which was not the case (now lines 102-105). This is now expressed more clearly in the methods (lines 374-375).

Second, I had a hard time understanding the way the data set was standardized. The idea of equalizing different categorical classifications into a binary “sensitive” vs “tolerant” seems oversimplified and potentially misleading.

Response: The scientific literature on correlated evolution is largely populated with analyses on binary characters that simplify biological traits that are complex in nature (Beaulieu et al., 2013 *Syst Biol*; Werner et al., 2014 *Nat Commun*; Werner et al., 2018 *Proc Natl Acad Sci USA*; Joly & Schoen, 2021 *Curr Biol*; Gardner and Organ 2021 *Syst Biol*; Pagel 1994 *Proc R Soc B: Biol Sci*). The particular use of the binary trait “sensitive” vs “tolerant” to drought in plants is not a new use and follows a recently published approach (Bowles et al, 2021 *Front Plant Sci*), as mentioned in the introduction (now lines 55-57) and in the methods (now lines 227-230). Therefore, in regard to the use of this binary character, I am confident that the manuscript meets state-of-the-art scientific standards.

The classifications in the original studies included observation of plants in different biomes, with particular local adaptations to prevalent climatic conditions, and using different approaches in their drought tolerant groupings. Thus, what is considered “fairly drought resistant” in one study could be equivalent to “medium” in another with no objective way to equalize both.

Response: The underlying rationale for standardization among the classification systems is based on the notion that all classification systems were built with the same purpose of comparing drought tolerance among plants in their biomes, and that each can be equally split in two levels. The standardization is needed to provide a global coverage that incorporates plants from different environments. This is now made clearer in the text (lines 230-234). For the particular classification systems mentioned by the reviewer, i.e. Dataset ID 92 and Dataset ID 98, each one uses a four-level classification. Therefore, the reviewer is right that, compared to other levels in Dataset ID 98, the level “fairly drought resistant” is likely the most equivalent to the level “medium” in Dataset ID 92. This was what the standardization has considered (lines 238-246); the two lower levels of Dataset ID 92 and of Dataset ID 98 were put together as a measure of drought sensitivity and the two higher levels of Dataset ID 92 and of Dataset ID 98 were put together as a measure of drought tolerance. Hence, although these may not be strictly equal, these assignments are considered in the manuscript to be suitable equivalences among classification systems based on the underlying rationale mentioned above. This is now stated in the text (lines 246-248). It is noteworthy that only a few plants were classified using Dataset ID 98, which corresponds to the New South Wales Plant Traits Database (lines 223-224). This database covers areas of the Australian continent that are not covered by Dataset ID 49 nor Dataset ID 92. Therefore, it was kept to incorporate plants living in arid and xeric environments of this part of the Globe. To make this clearer, this information has been added to the text (lines 224-226). Finally, the point raised by the reviewer falls within the scope of data uncertainty. As described in the text (now lines 355-363), and acknowledged by Reviewer 2 as “an admirable job”, the sensitivity analysis performed in my study showed that the main conclusions were robust in relation to data uncertainty (lines 86-89 and 118-119, Supporting Table S1 and Fig S1).

Similarly, the inclusion of facultative mycorrhizal and non mycorrhizal plants in the same “naked root” classification was poorly justified, particularly considering that the facultative stage can be meaningful in the understanding of mycorrhizal transitions in plant evolution (see Maherali et al 2017 *American Naturalist*).

Response: As expressed in the methods (now line 263), the mycorrhizal trait assignment is a matter of debate, with many opinion articles published in recent years. The debate is particularly disputed around the facultatively mycorrhizal plants. This is now made clear in the text (lines 263-264). In my manuscript, the inclusion of facultatively AM and non-mycorrhizal plants in the “naked root” strategy was only done when using the FungalRoot dataset as reference (now lines 284-287), but not when using the Bueno dataset (for which the naked root strategy only included non-mycorrhizal plants; now lines 292-295). I recognized that the abstract and the legends of some figures and tables may have been less clear on this matter, and this is now revised to correct it (line 12 and legends of Fig 1, Table 1, Supporting Fig S1, Fig S2, Table S3, and Table S5). The paper mentioned by the reviewer is certainly relevant to the present manuscript and is now considered in the text (lines 20-21; I believe the reviewer meant Maherali et al 2016 *American Naturalist*, not 2017, as there is no paper in 2017 by this author in this journal). However, as this aspect is a matter of debate, we should be clear in that the first author of the paper cited by the reviewer (Hafiz Maherali) has teamed up with the authors of the Bueno dataset to dispute the authors of the FungalRoot dataset (see Bueno et al 2019 *New Phytol*, 60 in my list of references). It is beyond the scope of the present manuscript to participate in this debate, and in my manuscript, I have considered both lines of thought by using their competing datasets. For the assignment of the naked root state using the highly cited FungalRoot reference (cited over 120 times in two years only), I prefer to use the considerations of the authors of this reference. These authors consider that many non-mycorrhizal and facultatively AM plants are habitat or nutritional specialists where mycorrhizas are irrelevant, as justified (now in lines 284-287). Furthermore, in my study, the naked root state functions as a negative control for obligate mycorrhizal symbiosis. Because facultatively AM plants have relatively lower frequencies of mycorrhizal associations (Soudzilovskaia et al 2018 *New Phytol*), and a lower frequency of mycorrhizal association is often considered an acceptable form of negative control to characterize the functions of mycorrhizas in the field (see the meta-analysis by Zhang et al 2018 *New Phytol*), this further justifies the inclusion of the facultatively AM plants in the predominantly naked root strategy. This additional justification was included in the text (lines 283-284 and 287-291). Finally, as for the previous concern, this particular aspect falls within the scope of data uncertainty. My sensitivity analysis on data uncertainty showed that each dataset version supported the overall main conclusions, i.e. all dataset versions supported the correlated evolution (Supporting Table S1) and all dataset versions unveiled robust differences among different mycorrhizal strategies in terms of their influence on drought adaptation evolution (Supporting Fig S1).

Finally, I believe the study lacks of insight with respect to recent literature, including important advances in root and fungal evolution. There are important developments in the evolution of root anatomy and morphology (Kong et al 2019 *Nat Comm*, Wang et al 2020 *Forest*, Valverde-Barrantes et al 2020 *New Phytologist*) that should be integrated in the paper, as part of the discussion.

Response: Unfortunately, I could not find the paper by Wang et al 2020 Forest based on the provided elements. If instead of 'Forest', the reviewer meant the journal 'Forests' of the Multidisciplinary Digital Publishing Institute, there is at least 18 papers published in 2020 whose first author name is Wang. Therefore, I could not consider this reference. However, for the two other suggested papers, these are now integrated in the discussion (lines 142-144).

In addition, the manuscript is imbued with jargon and vague points. Statements like "The best fitted HMMs revealed that this correlated evolution was consistently influenced by an unobserved (hidden) phylogenetic factor with two state levels (Supporting Table S1), which has either promoted or constrained the speed of evolution of the observed characters drought adaptation and mycorrhizal strategy" (l 90-92) are too general to make a clear point about the conclusions in the paper.

Response: The mentioned statement helps to clarify concerns raised by the reviewer on the influence of unobserved factors (such as diversification) on the evolution of the observed characters. Therefore, the statement must be kept in the text. However, to help non-specialists understand the meaning of this statement, a further clarification was added to the text (lines 95-100).

I recommend the author to be more specific in the analysis, targeting particular lineages and clarify how the transitions between drought tolerance stages influence mycorrhizal affiliation and evolution rates in both groups.

Response: The recommendation to target particular lineages is vague, tricky, and diverts from the scope of the manuscript (lines 68-70). The scope was to deliver a global-scale analysis on a correlated evolution. The recommendation is vague because I do not know which particular lineages should be targeted and why. It is tricky because the mycorrhizal strategy is highly conserved within lineages (Brundrett & Tedersoo, 2018 New Phytol; see also Fig 1a of the manuscript). For example, if I would target the monocot lineage because it is the largest in my dataset, this lineage does not have extant species with an ecto- or ericoid mycorrhizal strategy (Fig 1a), while this mycorrhizal strategy state was the one associated with the fastest transitions between drought adaptations (Fig. 1c). For other lineages, they either have only one mycorrhizal strategy (e.g. Ericales; Fig 1a) or have different mycorrhizal strategies extremely imbalanced (e.g. Fagales; Fig 1a), rendering impossible to properly test the hypothesis of correlated evolution. Therefore, this recommendation does not help to achieve the manuscript's objective.

Also, it will be important to look into the biogeography, and the ancestral reconstruction of the transitions, to have a better idea how these changes occurred in the past.

Response: The recommendation to look into biogeography would lead to a major fragmentation of the current data, reducing the power of the analysis. As stated in the legend of Fig. 1, and shown in Supporting Data S2, only 65% of the 1,638 plants have data available on geographical occurrences. Therefore, I would be limited to look into the biogeography of only approximately 1,000 plants (or only 500 for dataset v5 or v6). When this number is further fragmented into eight biogeographical realms, these fragmentations generate small imbalanced sub-sets. I am afraid that this recommendation disregards data limitations. Furthermore, I was not sure why the reviewer has requested "the ancestral reconstruction of the transitions, to have a better idea how

these changes occurred in the past". The text expresses that the estimated transitions occurred over the past 325My of evolution (e.g. lines 53 and 108), and these are shown in Fig 1c, Table 1, and Supporting Fig S1 and Supporting Table S5. Therefore, the study already shows the ancestral reconstruction of the transitions. To make this clearer, a statement was added to the text (lines 352-353).

As it is, I do not think the paper is ready for publication. I will be willing to make a more informed commenting on the paper, once this basic issues are addressed.

Response: I hope the reviewer can understand the need to keep the manuscript's focus and robustness, and that my responses and revisions address concerns and help to reach a fair, evidence-based decision.

Reviewer #2 (Remarks to the Author):

Summary

This manuscript provides evidence for a macroevolutionary correlation between symbiotic type and drought resistance in seed plants. The author uses discrete character Markov models to test for evolutionary correlation between the two characters. They analyze several datasets with alternative character codings and phylogenetic trees to test the sensitivity of their results. They find evidence of dependent evolution and suggest that symbiotic type is highly influential in drought resistance evolution. For my part, I am primarily knowledgeable about the methodological approach used in this study and so will focus comments primarily in that area. Overall, I believe this manuscript makes a novel contribution to a growing literature of correlated evolution and there are few shortcomings in the methodological approach taken here.

Response: I am thankful to the reviewer for recognizing the novel contribution of the research documented in the manuscript. The mentioned shortcomings are now addressed point-by-point below.

Major Comments

The author does an admirable job in examining the sensitivity of their results to different datasets. However, I would like to see the uncertainty in the parameter estimates for these datasets and best fitting models. My reasoning is that the results of any correlated evolution analysis are dependent on the rate estimates. Although the differences between transitions in drought tolerance depending on the mycorrhizal association state are large, the variance in those estimates could be insightful for determining whether we should trust these seemingly large differences. I believe the most straight forward way to estimate a measure of parameter estimation uncertainty would be to conduct a parametric bootstrap for your best fitting models. This would mean simulating data under your best fit model several hundred times and refitting the model under its own simulated data. You would then on average expect the same parameters to be found as those simulated under, but there will be some amount of uncertainty in those

estimates. That uncertainty/variance could then be reported as standard error around the original estimates and give a general idea of the robustness of them.

Another useful addition could be the conducting of power analysis ala Boettiger et al. (2012). This would be a similar procedure as the parametric bootstrap, but with the explicit purpose of contrasting models against one another. I.e., does this dataset have the power to detect correlated evolution instead of independent evolution. I do not think that both of these analyses need to be considered, but at least some measure of power or uncertainty is necessary to be included. My preference would be to conduct a parametric bootstrap so that there is some measure of uncertainty around the parameter estimates because the AIC indicates fairly strong evidence of the correlated model that power is unlikely to be an issue.

Response: I am thankful to the reviewer for appreciating my effort to examine the sensitivity of results in relation different datasets. The reviewer's comment on the lack of uncertainty around the estimated parameters is a valid and constructive criticism that has helped to improve the robustness of the analysis. Also, I am thankful to the reviewer for suggesting feasible approaches to help resolve this shortcoming. To provide confidence intervals of the estimated parameters, I have opted however to use an approach available in the corHMM package, which determines confidence intervals for the estimated parameters using the R package dentist. This approach consists of sampling points (set to 5,000 in my analysis) around a specified distance from the maximum likelihood estimates, and is considered a better way to estimate uncertainty over other approaches because it is theoretically better in detecting likelihood ridges that may not come out in a parametric bootstrap approach (James Boyko personal comments; <https://github.com/bomeara/dentist>). The resulting confidence intervals are now provided in Fig 1, Table 1, Supporting Fig S1, and Supporting Table S5. The respective R codes and output files were added to the Supporting Information. The new results on confidence intervals have led to major revisions in the abstract (lines 4-15; statement on drought adaptation effects on mycorrhizal strategy evolution was removed due to estimate uncertainty), introduction (line 75), results (lines 118-131), and methods (lines 363-369). In particular, the cases of weak estimates were identified and are now described in detail in the text (lines 118-131). Although this did not alter the main conclusions of the study, it led to a major revision on the differences between drought adaptation states in terms of their influence on the estimated rates of transitions among mycorrhizal strategies (lines 125-130). I hope that with these newly determined confidence intervals for the estimated parameters, together with the previously determined phylogenetic imbalance ratio (lines 102-105; which is a measure of whether the phylogenetic dataset contain sufficient information to infer correlated evolution; Gardner and Organ 2021 Syst Biol), the reviewer can agree that the analysis is sufficiently robust to test the two hypotheses of the manuscript.

Minor Comments

Line 6: plant tolerate should be tolerance

Response: This is now corrected in the text (line 6).

Line 11: remove “that”

Response: The word was removed from the text (line 11).

Line 141: “is markedly shaped by host-microbe associations”. Although this is true given your results, the evidence of hidden rate categories suggest that host-microbe interactions are not the only factor influencing the evolution of drought tolerance. It is worth highlighting the uncertainty of this finding and that there are other mechanisms at play shaping drought tolerance.

Response: To tone-down the statement, the sentence “markedly shaped by host-microbe associations” was replaced by the sentence “host-microbe associations can contribute to shape” (now lines 147-148). In addition, statements were tone-down in other parts of the manuscript (line 13 and lines 130-133). That said, the results do show that, regardless of the hidden rate category and dataset version, the EEM strategy promotes the rate of transitions between drought adaptation states, compared with the naked root strategy (Fig 1c, Table 1, and Supporting Fig S1). Nevertheless, it is correct that an unobserved factor is at play shaping the correlated evolution, and this is made clear in the results (lines 113-130) and now highlighted in the discussion (line 142 and 149).

References

Boettiger C., Coop G., Ralph P. 2012. Is Your Phylogeny Informative? Measuring the Power of Comparative Methods. *Evolution*. 66:2240–2251.

Response: this reference is now considered in the text (lines 372-373).

REVIEWERS' COMMENTS:

Reviewer #1 (Remarks to the Author):

The authors did a excellent job addressing the remarks suggested during the review process. I am satisfy with the explanations provided on the issues raised during the review, and there is a great improvement in terms of the quality and clarity of the manuscript.

My only observation for this new version is that the paper will benefit from a better description of these transitions among plant groups, that will illustrate these transitions. So far, the study described the results in terms of a number of transitions but does not explain what this means in terms of the actual groups that transitioned along the phylogeny. For instance, Gymnosperms showed two clades adapted to alternative mycorrhizal status, and within each clade there is an obvious trend of subgroups to be either drought sensitive or drought tolerant. However, for the reader there is no information about what does that mean. Are Cupressaceae more drought tolerant than Araucariaceae? I understand that explaining in detail the entire phylogeny is beyond the scope of this manuscript, but I do believe the paper will be more impactful if some iconic groups associated with drought tolerance (Poales, Caryophyllales, Gymnosperms, Arecales) are used to illustrate these transitions. Similarly, some groups, like Magnoliids, seems unable to acquire drought tolerance despite their success in wet tropical lands, why? This is a group with remarkable dependency on AM fungi that may be worth to discuss.

In summary, my comments on focusing on particular groups did not intend to decrease the sample pool (or diminish the analysis done) but actually make more accessible the results to the general public.

I hope the authors receive my comments as constructive.

Reviewer #2 (Remarks to the Author):

Major Comments

Having previously reviewed this manuscript my overall thoughts have remained unchanged as I believe this manuscript makes a novel contribution to a growing literature of correlated evolution. This subsequent review will then focus on how well the author addressed the methodological shortcomings I had previously outlined.

Previously, my main concern with this manuscript was whether confidence intervals (CIs) around the parameter estimates used to estimate correlation between symbiotic type and drought resistance would alter the author's conclusions. I am happy to see that indeed the main conclusions of the paper were unchanged by these newly estimated CIs and that the author has altered the manuscript where the parameter uncertainty was most relevant (lines 118-131). I also agree with the author that there was no need to necessarily conduct a parametric bootstrap if other methods (Dentist is used here) can adequately describe confidence intervals.

Minor Comments

Line 28: Sentence seems a little unclear. I'm not sure what is meant by "regardless of climate or vegetation type."

Line 105: It may be relevant to cite work which has discussed the issues of detecting correlation between discrete characters so that your readers can know why this is an issue which needs correcting in the first place. Specifically "Maddison, Wayne P., and Richard G. FitzJohn. "The unsolved challenge to phylogenetic correlation tests for categorical characters." *Systematic biology* 64.1 (2015): 127-

136." and "Boyko, James D., and Jeremy M. Beaulieu. "Reducing the biases in false correlations between discrete characters." *Systematic Biology* (2022)."

REVIEWERS' COMMENTS:

Reviewer #1 (Remarks to the Author):

The authors did a excellent job addressing the remarks suggested during the review process. I am satisfy with the explanations provided on the issues raised during the review, and there is a great improvement in terms of the quality and clarity of the manuscript.

My only observation for this new version is that the paper will benefit from a better description of these transitions among plant groups, that will illustrate these transitions. So far, the study described the results in terms of a number of transitions but does not explain what this means in terms of the actual groups that transitioned along the phylogeny. For instance, Gymnosperms showed two clades adapted to alternative mycorrhizal status, and within each clade there is an obvious trend of subgroups to be either drought sensitive or drought tolerant. However, for the reader there is no information about what does that mean. Are Cupressaceae more drought tolerant than Araucariaceae? I understand that explaining in detail the entire phylogeny is beyond the scope of this manuscript, but I do believe the paper will be more impactful if some iconic groups associated with drought tolerance (Poales, Caryophyllales, Gymnosperms, Arecales) are used to illustrate these transitions. Similarly, some groups, like Magnoliids, seems unable to acquire drought tolerance despite their success in wet tropical lands, why? This is a group with remarkable dependency on AM fungi that may be worth to discuss.

In summary, my comments on focusing on particular groups did not intend to decrease the sample pool (or diminish the analysis done) but actually make more accessible the results to the general public.

I hope the authors receive my comments as constructive.

Response: To address the reviewer's suggestion, I have added a new paragraph to the results that illustrates the major taxonomic lineages underlying the different mycorrhizal strategy groups, while describing the influence of these groups on the gains and losses of drought tolerance (lines 136-148). This includes references to the Fagales, Malpighiales, Pinales (Gymnosperms), Ericales, Rosales, Poales (monocots), Asterales, Fabales, Lamiales,

Caryophyllales, Alismatales (monocots), Brassicales, and Lamiales. In addition, the plant illustrations in Fig. 1b were revised to specifically illustrate these iconic taxonomic groups, making the results more accessible to the general public.

Reviewer #2 (Remarks to the Author):

Major Comments

Having previously reviewed this manuscript my overall thoughts have remained unchanged as I believe this manuscript makes a novel contribution to a growing literature of correlated evolution. This subsequent review will then focus on how well the author addressed the methodological shortcomings I had previously outlined.

Previously, my main concern with this manuscript was whether confidence intervals (CIs) around the parameter estimates used to estimate correlation between symbiotic type and drought resistance would alter the author's conclusions. I am happy to see that indeed the main conclusions of the paper were unchanged by these newly estimated CIs and that the author has altered the manuscript where the parameter uncertainty was most relevant (lines 118-131). I also agree with the author that there was no need to necessarily conduct a parametric bootstrap if other methods (Dentist is used here) can adequately describe confidence intervals.

Minor Comments

Line 28: Sentence seems a little unclear. I'm not sure what is meant by "regardless of climate or vegetation type."
Response: The sentence was changed to "in different climates and vegetation types" (lines 33-34).

Line 105: It may be relevant to cite work which has discussed the issues of detecting correlation between discrete characters so that your readers can know why this is an issue which needs correcting in the first place. Specifically "Maddison, Wayne P., and Richard G. FitzJohn. "The unsolved challenge to phylogenetic correlation tests for categorical characters." *Systematic biology* 64.1 (2015): 127-136." and "Boyko, James D., and Jeremy M. Beaulieu. "Reducing the biases in false correlations between discrete characters." *Systematic Biology* (2022)."
Response: The recommended references are now cited in the text (lines 115-366).